

# The effect of different landslide mapping approaches on the geomorphological assessment of landslide hazard

Marco Donnini[1], Francesco Bucci*[1], Michele Santangelo[1], Mauro Cardinali[1], Paola Reichenbach[1]

[1] Consiglio Nazionale delle Ricerche (CNR), Istituto di Ricerca per la Protezione Idrogeologica (IRPI), Perugia, Italy

* Corresponding author: francesco.bucci@cnr.it

## Abstract

Despite various methodologies proposed in the last decades, there are still no standard for estimating landslide hazard. Consequently, practical applications for territorial management have to assimilate in a single cartography information obtained at local level with different methods, with negative consequences on the quality of derived products. Here we proposed a new methodology - based on well-established hazard matrices - to assess landslide hazard, which starts from a landslide inventory, and introduces a new method for estimating the landslides frequency. We apply this new method to three landslide inventories compiled with increasing detail. They are: (i) a basic-historical inventory, (ii) a generational-historical inventory (a detailed version of a simple historical inventory), (iii) and a composite multi-temporal inventory (which includes the generational-historical inventory plus the multi-temporal inventory). Results are then compared each other, and to independent measures from Persistent Scatterer Interferometry. Our results highlight the importance to base landslide hazard analysis on a generational-historical inventory that adequately characterizes the complexity of landslide clusters, whereas indicate that multi-temporal mapping is not decisive for the purpose. Overall, our procedure puts landslide mapping back at the center of the hazard assessment chain, raising questions on the reliability and availability of landslide inventory maps.

## 1. Introduction

Landslide hazard can be defined as the probability that in a given area a landslide of a given intensity (or magnitude) occurs in a given period of time (see e.g. *Varnes, 1984*; *Guzzetti et al., 1999*). Various methodologies have been proposed for landslide hazard estimation, including qualitative matrix (*BUWAL, 1999*), geomorphological approach (*Canuti & Casagli, 1996*), quantitative statistical (*Guzzetti et al. 2007*) or a combination of multiple and mixed methods (*Trigila et al., 2018*). All methodologies are based on landslide inventories and consider parameters such as type of movement and state of activity. Each of these methods has strengths and weaknesses.



The qualitative matrix methods consider the landslide polygons recorded in an inventory and are based on a different number of parameters spanning from one (e.g.,state of activity), two (e.g. state of activity and type of movement), or more (e.g. spatial probability of occurrence, the estimated velocity and the size of landslides (e.g. *Cardinali et al., 2002*). The qualitative matrix methods offer the advantage of being replicable and grounded in simplified schemas, but as main limitation classify only areas already affected by landslides.

The geomorphological methods classify the slopes based on geomorphological and geological characteristics (e.g. ongoing landslide phenomena, morphological indications of instability, lithologies with a high propensity to landslides). The advantage of these methods is the classification of the entire investigated territory, but they suffer from an inherent subjectivity and lack quantitative information about the temporal frequency and the magnitude of the hazard.

Quantitative statistical methods determine the weight of the factors that contribute to instability (e.g. steepness, lithology, land use), through bivariate or multivariate analysis. The methods have the advantage of objectivity and reproducibility of a spatially continuous determination of landslide susceptibility but present the limitation of scarcity of data on landslide temporal frequency and magnitude. As a result, many approaches do not incorporate such information in the hazard evaluation, leading to mistakenly using hazard and susceptibility terms as synonyms (*Reichenbach et al., 2018, Corominas et al., 2023*).

However, despite a wide literature (see *Tyagi et al., 2022* and reference therein for an exhaustive list), there are still no standard for estimating landslide hazard, and practical applications for territorial management often have to assimilate in a single cartography information obtained at the local level with different methods. Such a process severely impacts the quality of derived cartographic products, which are often characterized by spatial inhomogeneities and questionable content, difficult to interpret and use.

*Cardinali at al., (2002)* proposed a methodology to assess landslide hazard that starts from mapping all the existing and past landslides within a given area, i.e. by elaborating a landslide inventory map. Successively, considering the observed changes in the landslide pattern and distribution, the authors suggest a method to deduce the possible slope evolution and the expected occurrence frequency. In the approach proposed by *Cardinali et al. (2002)*, the landslides were first mapped in a historical inventory map showing the distribution of past landslides, then elaborating a multi-temporal inventory map showing the occurrence of more recent landslides over a period of about 60 years in the study area, by using a set of stereoscopic aerial images of different periods. Although empirical and to some extent subjective, the method proposed by *Cardinali et al. (2002)* can provide reasonable estimates of landslide hazard in its three dimensions: (i) expected magnitude as a proxy of estimated velocity and volume; (ii) spatial occurrence expressed as evolutionary scenarios of existing landslides, and (iii) frequency of landslides as obtained from the multitemporal landslide mapping. However, this does not express the frequency of landslides older than the last 60 years, which instead - very importantly - represent





almost the entire landslide area of a territory. This circumstance poses issues
regarding the representativeness of the landslide hazard obtained by this method.
In this paper we approach these issues building on the method proposed by Cardinali
et al. (2002) - hereinafter called *original method* - introducing a new method for
estimating the frequency of all landslides (slow, fast and rapid) recognized over a
territory.
We apply this new method to three landslide inventories available for the test area
(Militello Rosmarino, NE Sicily, Southern Italy), compiled by the same authors with
increasing detail. They are: (i) a *basic-historical* inventory, (ii) a *generational-historical*
inventory (which is a detailed version of a simple historical inventory), (iii) and a
*composite multi-temporal* inventory (which includes the *generational-historical*
inventory plus the *multi-temporal* inventory). The goal of this study is to investigate
how the different landslide inventory maps may influence the assessment of landslide
hazard and establish under which conditions the estimation of landslide hazard can
be based only on information derived from a historical landslide inventory.
Compared to the original method, the new method introduce three main novelty to the
hazard assessment chain: (i) the preliminary evaluation of the informative content of
the available landslide inventory maps; (ii) the generational criterion in the landslide
frequency estimation; (iii) the use of the information derived from the ground motion
time series as obtained through the persistent scatterers (PS) technique (see e.g.
Ferretti et al. 2000; 2001; Colesanti et al., 2003; Bianchini et al., 2015) to compare the
hazard estimation to independent measures. While the first two methodological
advances concern all landslides, the comparison with PS data concerns only slow-
moving landslides. However, it is worth mentioning that slow-moving landslides
(typically slides and complex landslides) are the most abundant, at local (about 80%
of total landslides number, according to this work), national and continental scale
(about 70% of total landslides number in Italy and Europe, according to *Trigila et al.,*
*2021* and *Herrera et al., 2017*) and for this reason they deserve a specific focus in this
contribution. Finally, we discuss the impact of our results on the landslide hazard
estimation of the investigated area, providing a scientifically rigorous methodological
framework potentially reproducible in any geomorphological context.

## 2. Study area

We chose as a study area the village of Militello Rosmarino (NE Sicily, Southern Italy)
and its surroundings (**Figure 1b**), located in the Nebrodi Mountains of the Messina
province, which are part of the Apennine-Maghrebian orogenic chain (e.g., *Bosellini et*
*al., 2017*). This region is characterized by a highly complex geological framework, with
outcrops of terrigenous, calcareous, and metamorphic rocks (e.g., *Cimino et al., 1998*;
*Cubito et al., 2005*; *Bianchini et al., 2015*; *Ruggieri, 2022*). The study area covers
approximately 2 km² and encompasses anthropic settlements bordered by agricultural
zones, forested areas, and semi-natural landscapes. The area is particularly
susceptible to landslides, including rockfalls, debris flows, complex landslides, and





both shallow and deep-seated slides. These phenomena are primarily driven by the
steep topography, the nature of the local lithologies and the morpho-structural setting,
and the occurrence of very intense seasonal rainfall events (e.g., *Mondini et al., 2011*;
*Ardizzone et al., 2012*; *Del Ventisette et al., 2012*; *Raspini et al., 2013*; *Donnini et al.,*
*2017*). **Figure 1c**, illustrates a section of the land cover/use map at a 1:10,000 scale
distributed by the Regional Authority (www.sitr.regione.sicilia.it) highlighting that the
eastern part of the study area exposes rock outcrops, while a north-south-oriented
broad riverbed delineates the eastern boundary. **Figure 1d**, which was generated
using the 1:100,000 scale lithological map of Italy by *Bucci et al. (2022)*, indicates the
predominance of carbonate rocks in the eastern sector of the study area, associated
elsewhere by siliciclastic formations such as sandstone, mudstone, and greywacke,
alongside clastic deposits.
**3. Data**
To evaluate the landslide hazard within the study area, different input data were used:
(i) a set of stereoscopic images (**Table 1)**, (ii) *basic-historical* landslide inventory map,
(iii) *generational-historical* landslide inventory map, (iv) *composite multi-temporal*
landslide inventory map, (v) the Persistent Scatterers (PS) derived by ERS, ENVISAT
and COSMO-SkyMed, (vi) the land cover/use map "*Carta dell'Uso del Suolo secondo*
*Corine Land Cover*" (**Figure 1c**), and (vii) the 1:10,000 topographic map "*Carta*
*Tecnica Regionale*" (CTR, base map of **Figure 1c** and **1d**).

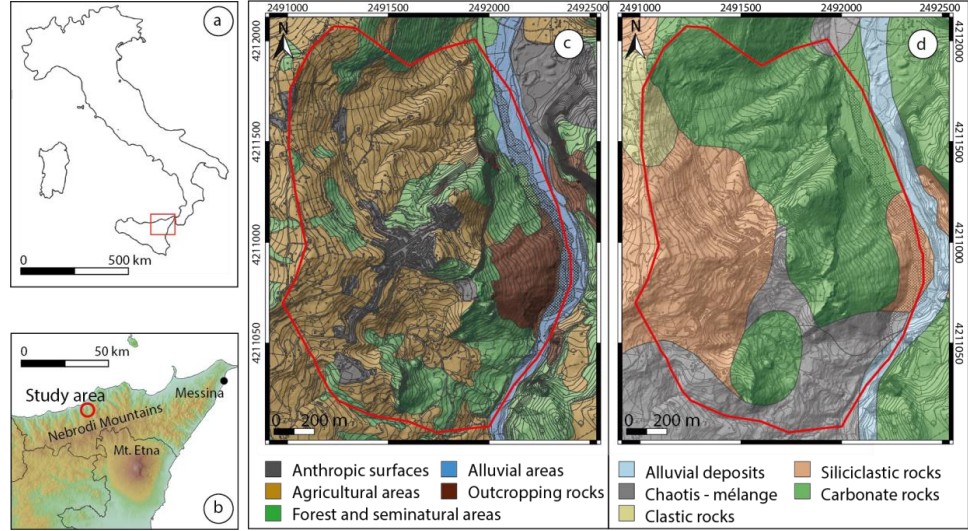

**Fig. 1** (a) Italy, (b) Study area, (c) land cover/use and topographic maps released by Regione
Siciliana at 1:10,000 scale (www.sitr.regione.sicilia.it), (d) lithology derived from the lithological
map of Italy at 1:100.000 scale (*Bucci et al., 2022*). In (c) and (d) the boundary of the study
area is highlighted in red. Base maps derived from 2m LiDAR DEM (www.sitr.regione.sicilia.it).




**Table 1** Aerial photographs used in this work. GAI - IGMI = *Gruppo Aeronautico Italiano* (Italian
Aeronautical Group) - *Istituto Geografico Militare Italiano* (Military Geographical Institute).
SITR = *Sistema Informativo Territoriale Regionale* (Regional Territorial Information System)

| Year | Period | Type | Format | Scale | Reference |
|------|--------|------|--------|-------|-----------|
| 1955 | Summer - Autumn | Panchromatic | *tiff | 1:33,000 | GAI - IGMI |
| 1977 | Autumn-Winter (November-December) | Black and white | *jpg | 1:18,000 | SITR |
| 1987 | Spring-Summer (May-June) | Coloured | *jpg | 1:10,000 | SITR |
| 1997 | Not specified | Black and white | *tiff | 1:20,000 | SITR |
| 2005 | Spring - Summer | Panchromatic | *tiff | 1:29,000 | IGMI |
| 2013 | Not specified | Coloured | *ecw | 1:10,000 | SITR |


### 3.1 Stereoscopic images

The stereoscopic aerial photographs of the study area were chosen considering a time
interval of ~10 years between each flight over the period 1955-2013 (Table 1).
The 1955 and 2005 images were acquired by IGMI (*Istituto Geografico Italiano*, Italian
Military Geographical Institute; *www.igmi.org*). Specifically, the 1955 images are from
GAI-IGMI (GAI=*Gruppo Aeronautico Italiano*, Italian Aeronautical Group). The 1977,
1987, 1997, and 2013 images were released by *Regione Siciliana* - SITR (*Sistema
Informativo Territoriale Regionale*, Regional Territorial Information System;
*www.sitr.regione.sicilia.it*) with authorization 2020-E-2851[1].

### 3.2 Landslide inventory maps

In this work we use three different landslide inventory maps, covering the same study
area with increasing detail:
(ii) *basic-historical* landslide inventory map,
(iii) *generational-historical* landslide inventory map,
(iv) *composite multi-temporal* landslide inventory map.
The *basic-historical* inventory map, the less detailed inventory, was obtained by the
interpretation of the aerial photographs sets acquired in 1955 and 2005. Both sets
were used because comparing the appearance of the landscape in different periods
can reduce uncertainty in the interpretation. Landslides mapped on each of the two
flights were classified as "pre-1955" and occurred in uncertain historical periods. The

---

[1] The images are owned by the Sicilian region released on 04/12/2020 with document No. 2020-E-2851 (*elemento di proprietà della Regione Siciliana ceduto in data 04/12/2020 al N. 2020-E-2851*)



interpretation was performed by using analogic "discussion" stereoscopes with lenses
and mirrors, with double zoom capability. Landslide polygons were reported visually
on a topographic base map at a scale of 1:25,000 (*Santangelo et al., 2015; Bucci et*
*al., 2016*).
The *generational-historical* inventory map was prepared through systematic visual
interpretation of the same images used for the *basic-historical* inventory and contains
detailed information on the overlapping of landslides of different generations (up to
three generations). In such contexts, the relative age of the landslides was assigned
based on morphological evidence and cross cutting relationships of the failures (*Bucci*
*et al., 2021; Ardizzone et al., 2022*). Such a generational classification approach is
therefore valid only within the same landslide cluster and not among clusters, defining
a cluster as a system of partially or totally overlapping landslides. The stereoscopic
analyses were performed in digital mode with the use of 3D vision glasses associated
with a dual-screen computer where StereoPhoto Maker4 (https://stereo.jpn.org) and
QGIS (www.qgis.org) were installed. Compared to the analogic stereoscopes
described earlier, StereoPhotoMaker allows for a continuous zoom up to 250%, which
corresponds to a scale <1:1000 on the aerial photographs used. Landslides were
reported directly on a 2m LiDAR DEM distributed by Regione Siciliana.
The *composite multi-temporal* inventory is based on the *generational-historical*
inventory and includes all the recent landslides occurred during the period ranging
from 1954 and 2013, covered by the six aerial photographs sets shown in **Table 1**. In
the multi-temporal landslide inventory map: landslides are classified as 1955, 1977,
1987, 1997, 2005, and 2013 if interpreted as occurred close to the image acquisition
time, otherwise they were classified as occurred in the interperiod 1955-1977, 1977-
1987, 1987-1997, 1997-2005, 2005-2013 based on the evidence of geomorphological
changes. Interpretation and mapping were carried out as for the *generational-historical*
inventory.
**Figure 2** shows the three landslide inventory maps, with landslides classified
according to their age (**Figures 2a, 2c, 2e**) and type (according to *Hungr et al., 2014;*
**Figures 2b, 2d, 2f** and **Section A.1** of the ancillary materials). **Figure 2b, 2d, 2f** show
that, when possible, for each landslide the deposit was mapped separately from its
source area (i.e. scarp, coded with an "x" preceding the type code). Visual inspection
of **Figure 2** shows that the most abundant landslide type is slide (*s*), and that rockfall
(*rf*) and rockfall areas (*rfa*) are present in the eastern sector of the study area where
outcropping rocks are present (**Figure 1c**).
Landslide number increases from the *basic-historical* inventory to the *composite multi-*
*temporal* inventory. Overall, **Figure 2** shows that the landslides that occurred after
1955 recognized by the multi-temporal landslide inventory map are a small percentage
of all landslides recognized in the other maps, and that they almost entirely fall into
areas already affected by previous mass movements. Such a trend is expected
considering that geomorphological historical inventories refer to time spans that are at
least three orders of magnitude larger than multi-temporal inventories.





**Fig. 2** *Basic-historical* (Pre-1955) landslide inventory map with landslides classified according to (a) their relative age, and (b) type. *Generational-historical* (Pre-1955) landslide inventory map with landslides classified according to (c) their relative age, and (d) type. *Composite multi-*



*temporal* landslide inventory map with landslides classified according to (e) their relative age,
and (f) type. Base map derived from 2m LiDAR DEM (www.sitr.regione.sicilia.it).

### 3.2.1 Informative content of landslide inventory maps


To evaluate the informative content of the *basic-historical* and *generational-historical*
inventories, we compared them with two recently compiled inventories from southern
Italy, i.e. Daunia and Val d'Agri (*Ardizzone et al., 2022; Bucci et al., 2021*), where
lithologies and landscapes similar to our study area occur (*Bucci et al., 2022*). We
consider two catchment-scale, less than 10 km2 wide, subsets of these inventories to
serve as comparison with the inventories over our study area, since their high level of
detail meets the standards required for catchment-scale hazard analysis (*Zumpano et
al., 2021*).
The results of the comparison are shown in **Table 2**. The comparative table is
composed by rows listing materials and methods used, and includes data on the
informative content of the inventories, such as the maximum number of landslide
generations recognized, the size of the smallest landslide and the adopted
cartographic scale (i.e. publication scale).
**Table 2** shows that, despite the similar scale of aerial photographs used for all
inventories (~1:30,000), the level of detail in photo-interpretative analysis varies
significantly, leading to different outcomes. For instance, the *basic-historical inventory*,
compiled using analog stereoscopes with a maximum 3× zoom, lacks the precision
achievable with digital stereoscopes that offer continuous zoom. A more detailed
analysis enables the identification of more landslides, including smaller ones, and a
greater number of landslides generations.
Additionally, high-resolution LiDAR-derived topography allows for more accurate
representation (i.e. position, size, shape, *Santangelo et al., 2015*) of small landslides
at scales larger than 1:10,000. **Table 2** further indicates that the *generational-historical
inventory* aligns more closely with reference inventories in qualitative metrics, whereas
the *basic-historical inventory* shows notable discrepancies.





**Table 2** Comparison of the informative content, materials and methods used for the examined
*basic-historical* and *generational-historical* inventories with those used as reference.

| | Reference inventories | | Examined inventories | |
| --- | --- | --- | --- | --- |
| | Historical Val d'Agri | Historical Daunia | *basic-historical* | *generational-historical* |
| Scale of aerial photos | 1:34,000 | 1:30,000 | 1:29,000 | 1:29,000 |
| Digital stereoscope | Yes | Yes | No | Yes |
| Continuous zoom | Yes | Yes | No | Yes |
| Observation scale at max zoom | 1:2,500 | 1:2,000 | 1:7,500 | 1:2,000 |
| Size of the study area (km2) | 5.7 | 7.7 | 2 | 2 |
| Total landslides number | 124 | 145 | 22 | 44 |
| % slow moving landslides | 63% | 78% | 78% | 79% |
| % rapid landslides | 9% | 22% | 9% | 5% |
| % fast moving landslides | 30% | 0.0% | 13% | 16% |
| % slow moving landslides area | 79% | 86% | 94% | 95% |
| % rapid landslides area | 10% | 14% | 1% | 1% |
| % fast landslides area | 11% | 0.0% | 5% | 4% |
| Landslide density (#lnds/Km2) | 20 | 19 | 11 | 22 |
| Max number of landslide generations in map | 4 | 4 | 2 | 3 |
| Drawing on HR lidar derived topography | Yes | Yes | No | Yes |
| Drawing scale | multiple | multiple | 1:15,000 | multiple |
| Size of the smallest element in map (m2) | 100 | 25 | 400 | 100 |
| Publication scale | 1:10,000 | 1:5,000 | 1:20,000 | 1:10,000 |
| Purpose of the study | application | application | knowledge | application |






### 3.3 Persistent scatterers

The persistent scatterers (PS) technique is widely used for detection of slow-moving landslides at medium-large scales (from 1:100,000 to 1:5,000, see e.g. *Fell et al., 2008*; *Cigna et al., 2013*) in urbanized and artificial areas where PS benchmarks are often abundant (*Bianchini et al., 2012*). However, high PS density values are also found in correspondence with rocky outcrops, cones and debris covers with absent or sparse vegetation (*Riddick et al., 2012*), which is a fairly widespread condition in our study area.

In this paper we used PS data, namely SAR data in C-band from ERS (observation period 1992-2001), and ENVISAT (observation period 2003-2010) satellites, that has been demonstrated to be a valuable tool for back-monitoring slow-moving landslides, with good accuracy (up to 1 mm/year) and maximum detectable movement of about 20 mm/year (*Hanssen, 2005*; *Cascini et al., 2010*; *Cigna et al., 2013*).

Moreover, we also used the PS derived from the SAR sensors in X-band of COSMO-SkyMed satellites (observation period 2011-2012), with higher spatial resolution and reduced revisiting time compared to the C-band satellites, allowing the identification of more recent and faster ground movements affecting small areas with improved precision (see *Bianchini et al., 2015* and references therein).

PS data were obtained as part of the DORIS (Ground Deformation Risk Scenarios: an advanced Assessment Service) project, an integrated Seventh Framework Program project of the European Commission (www.doris-project.eu).

Overall, PS data cover a period of 20 years without significant interruptions offering a continuous time series of ground motion over the study area.

### 3.4 Land cover/use and topographic maps

The land cover/use map "*Carta dell'Uso del Suolo secondo Corine Land Cover*" (**Figure 1c**) was defined according to the criteria of the CORINE LAND COVER, (*Corine Land Cover, 2021*) and was released in vector format by the Regione Siciliana at 1:10,000 scale (www.sitr.regione.sicilia.it), as the topographic map ("*Carta Tecnica Regionale*" at 1:10.000 scale, base map of **Figure 1c** and **1d**).

## 4. Method

In this work we use three different landslide inventory maps, (see **Section 3.2**), to define landslide hazard by applying a modified version of the approach proposed by *Cardinali et al. (2002)* (**Figure 3**), a heuristic approach that evaluates landslide hazard through the definition of scenarios named Landslide Hazard Zones (*LHZs*), defined as areas of possible (or probable) evolution of existing landslides with similar characteristics (i.e. of type, volume, depth, and velocity; **Section A.2** of the ancillary materials). While *Cardinali et al. (2002)* did not classify pre-1955 landslides by generational criteria in the *composite multi-temporal* inventory map, the methodology



proposed here includes such a classification. The new method is applied to each of
the three inventories and results are compared with the original method proposed by
*Cardinali et al. (2002)*.

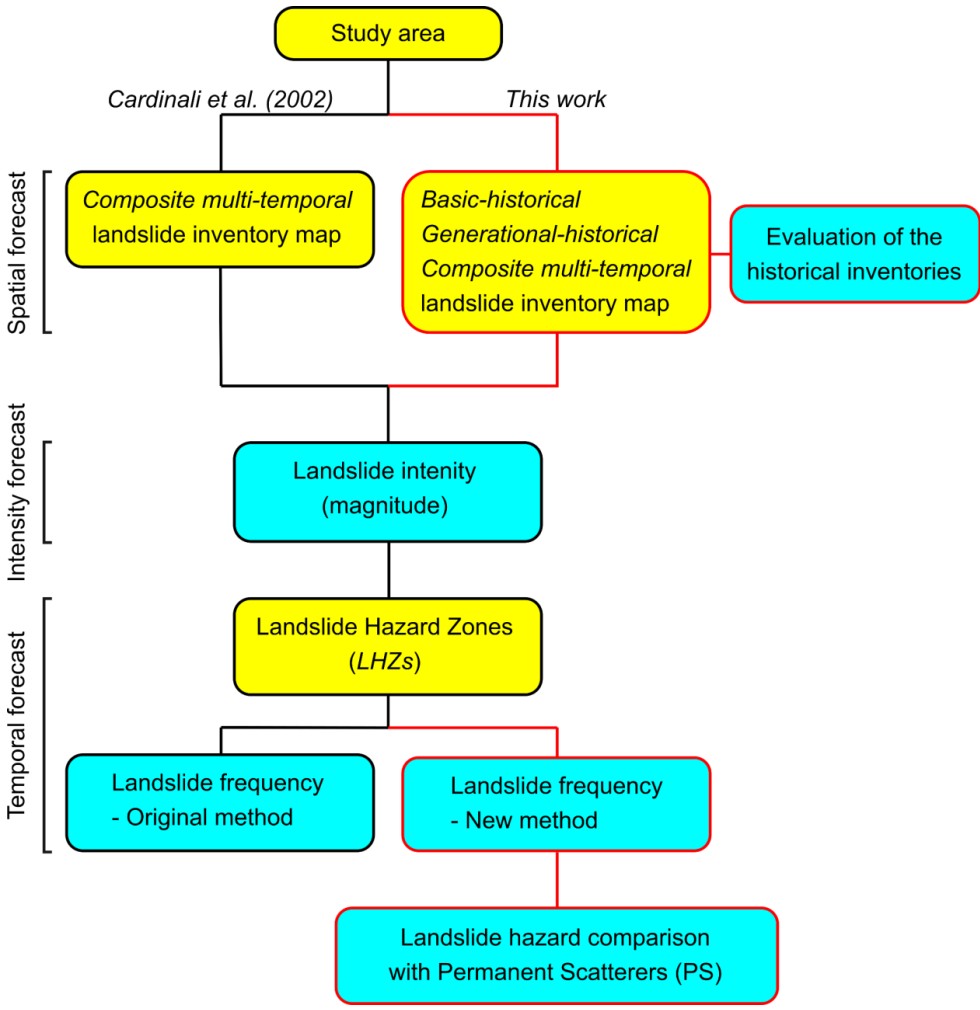

**Fig. 3** Flow-chart of the methodology proposed in the paper. Black lines/polygons represent
the logical processes already presented in *Cardinali et al. (2002)*, while red lines/polygons
represent those proposed in this paper. Yellow boxes represent the procedures where
mapping is involved, cyan boxes represent those applied within the mapped areas.
The black and red lines/polygons of **Figure 3** represent the logical processes already
presented in *Cardinali et al. (2002)* (black lines/polygons) and those proposed in this
paper (red lines/polygons). The yellow boxes represent the procedures where
mapping was involved, whether for the study area definition (**Section 2**), landslide
inventories preparation (**Section 3.2**), and Landslide Hazard Zones (*LHZs*)



delimitation. The cyan boxes represent the procedures applied within the defined areas, i.e. (i) the evaluation of the landslide inventory maps, (ii) the landslide intensity definition, (iii) the landslide frequency counting, (iv) the landslide hazard assessment, (v) and the landslide and *LHZs* comparison with the persistent scatterers information. Each of the previous steps is discussed below.

### 4.1 Landslide intensity definition

According to *Cardinali et al. (2002)*, landslide intensity as a proxy of landslides destructive capacity can be considered a function of landslide volume and expected velocity. It can be expressed through a positional index (**Figure 4**) in which, starting from the left, the first digit refers to the estimate of the landslide volume discretized in four classes (from 1 to 4), and the second digit expresses the expected landslide velocity discretized in three classes (1: slow, 2: rapid, 3: fast). Slides and slide-flows belong to slow landslides; debris-flows belong to rapid landslides; and rockfalls are considered fast landslides. Flows can be considered both slow and rapid, to take into account the high expected variability of the flow velocity. The magnitude of rockfall (*rf*) and rockfall areas (*rfa*) is measured by the volume of the maximum expected boulder involved, which can be estimated through images interpretation and/or field survey. **Figure 4** shows the grouping and ranking of all the possible values of landslide intensity in four classes: low, medium, high, very high (**Section A.3** of the ancillary materials).

| Landslide volume (m³) | Expected landslide velocity | | |
|---|---|---|---|
| | Slow landslides | Rapid landslides | Fast landslides |
| < 0.001 | | | Low (13) |
| < 0.5 | | | Medium (23) |
| > 0.5 | | | High (33) |
| > 500 | | Low (12) | High (33) |
| 500 - 10,000 | Low (11) | Medium (22) | High (33) |
| 10,000 - 500,000 | Medium (21) | High (32) | Very high (43) |
| > 500,000 | High (31) | Very high (42) | |
| >> 500,000 | Very high (41) | | |

**Fig. 4** Definition of landslide intensity, modified from *Cardinali et al. (2002).*

### 4.2 Landslide frequency counting

According to *Cardinali et al. (2002)*, landslide frequency is defined in each *LHZ* by counting (i) the number of periods of the multi-temporal inventory in which at least one landslide has been recognised, (ii) landslides that occurred before the first period (i.e. pre-1955 in this study) as a single time step, even if belonging to different generations. So, for instance, if for a given intensity in a given *LHZ* there are two landslides pre-1955, or two landslides in 1976 (**Figure 5d**), the overall frequency in both the cases





would be 1 (**Figure 5f**), while the frequency would be 2 in the case of a *LHZ* containing
one landslide in 1976 and one landslide in 2005 (**Figures 5d**, **5f**).
Here, we extend the count of the number of events also to landslides that occurred
before the first flight available, i.e., during an undefined time period. In the same
conditions as the previous example, the overall frequency would be 1 only for the *LHZ*
containing the two landslides occurring in 1976, in a single time step, while would be
2 in the other two cases (**Figures 5d**, **5e**).
The choice not to count all the landslides of each temporal layer except that
antecedent the first flight available refers to the fact that it is very unlikely that
landslides occurred before the first image belong to the same event, as opposed to
the landslides recorded in the multi-temporal time steps.
Inspection of Figure 5 shows that, compared to the original method, the new method
proposed in this work assigns a higher frequency value to a given *LHZ* only where
multiple pre-1955 landslides were recognised, regardless of their generation.
Finally, the new method always considers the *LHZs* related to rockfall and rockfall area
with the highest frequency, thus recognizing the high recurrence, often seasonal, of
such events that contribute to maintaining the morphological freshness of source
areas and talus.
Similarly to the intensity classes, the frequency value enters the positional index as
the first digit on the left: 1 (low frequency - one event), 2 (medium frequency - two
events), 3 (high frequency - three events), and 4 (very high frequency - four or more
events).






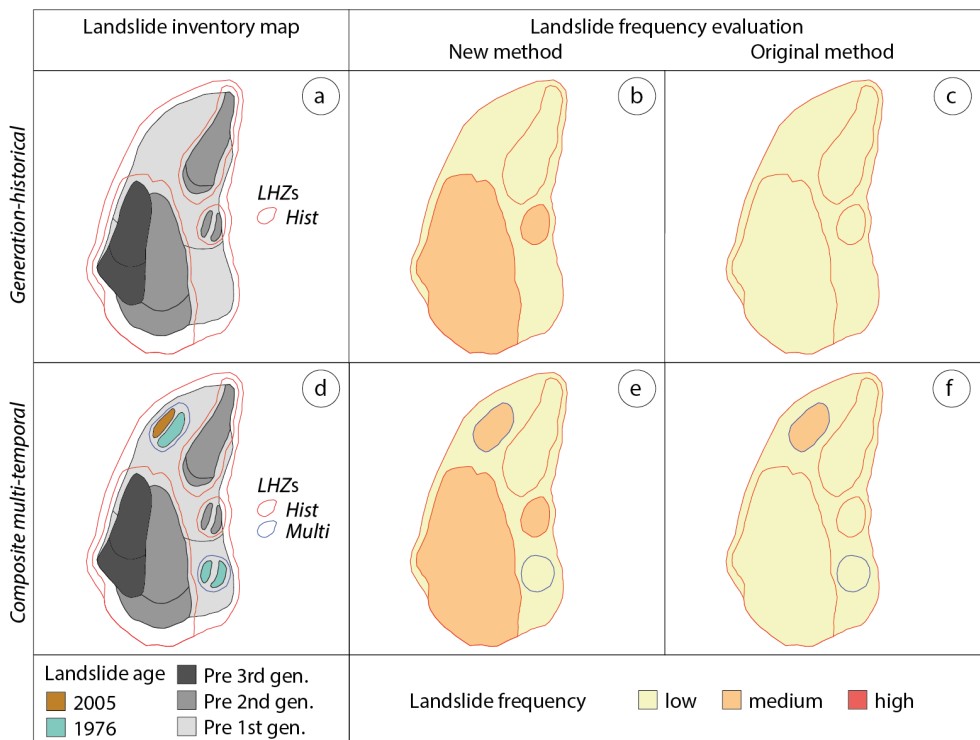


**Fig. 5** Comparison of the original method and the new method of landslide frequency counting applied to a hypothetical landslide cluster for the *composite multi-temporal* (d, e, f) and the *generational-historical* (a, b, c) inventories.

## 4.3 Landslide hazard assessment

**Table 3** shows the hazard index composed by three digits. From the left, the first digit refers to frequency, the second to the magnitude and the third to the velocity (these last two expressing the intensity). Therefore, a *LHZ* with a hazard index of 321 is characterized by a high frequency (3) and a medium intensity (21). The theoretical values of the hazard index of **Table 3** can then be ranked and grouped in hazard classes according to several criteria that should be discussed with decision makers. Since from this point on the procedure will follow the same approach described by Cardinali et al., (2002), grouping the indices in hazard classes is not proposed.

**Table 3** Definition of *LHZ* hazard class, modified from *Cardinali et al. (2002)*

| | | Intensity | | | |
|---|---|---|---|---|---|
| | | 11/12/13 (low) | 21/22/23 (medium) | 31/32/33 (high) | 41/42/43 (very high) |
| Frequency | 100 (low) | 111/121/113 | 121/122/123 | 131/132/133 | 141/142/143 |
| | 200 (medium) | 211/212/213 | 221/222/223 | 231/232/233 | 241/242/243 |





| | 300 (high) | 311/312/313 | 321/322/323 | 331/332/333 | 341/342/343 |
| --- | --- | --- | --- | --- | --- |
| | 400 (very high) | 411/412/413 | 421/422/423 | 431/432/433 | 441/442/443 |


### 4.4 Landslides and *LHZs* comparison with Persistent Scatterers (PS)


In the study area, PS data were treated as follows: (i) overlapped to the land use/cover
map to verify their actual coverage; (ii) classified based on the absolute values of their
average velocity (mm/y) to avoid negative values and to distinguish stable (close to 0)
and unstable points, independently from the acquisition orbit; (iii) used, through a
kernel density estimation, to create a density map based on the number of points in a
location, weighted by the velocity attribute field, to highlight clusters of moving points
to be compared with the mapped landslides; (iv) used within individual *LHZ* of slow
moving landslides to generate contour layers based on the velocity attribute field.
Where available, such information was compared to the frequency component of the
hazard matrix associated with *LHZ* of slow-moving landslides, to verify spatial
correlation between *LHZ* with frequency larger than 1 and unstable areas identified by
PS data.

## 5. Results

### 5.1 Landslide intensity evaluation

The landslide intensity is shown in **Figures 6** for the *basic-historical inventory*, in
**Figures 7** for the *generational-historical inventory* and in **Figure 8** for the *composite
multi-temporal inventory*. **Figure 9** shows a synthesis of the differences in the number
of landslides recorded in the three inventories.
In **Figures 6-8**, the first four columns display landslides (filled polygons) overlaid on
their corresponding *LHZs* (outlined polygons), categorized by movement velocity —
slow, rapid, and fast — and classified by intensity (low, medium, high, very high; see
**Section 4.2**). The last column presents the *LHZs* with all intensity levels overlapped.
As illustrated in **Figures 6-8**, in all the inventories, *LHZs* for slow landslides are the
most abundant and vary significantly in size. In contrast, *LHZs* for fast and rapid
landslides are less frequent and cover smaller areas.
Comparison of **Figures 7, 8** shows that the contribution of the multi-temporal
component of the *composite multi-temporal inventory* is limited to low intensity and low
velocity landslides.
A closer examination reveals that slow landslides (**Figures 6b–d**, and **Figures 9a-d**)
are less represented in the *basic-historical inventory* compared to the *generational-
historical* (**Figure 7b-d**) and the *composite multi-temporal inventory* (**Figures 8b–d**).
Furthermore, slow landslides of low intensity are predominantly identified within the
*composite multi-temporal inventory* than in the remaining inventories (**Figures 6a, 7a,**





and **8a**, **9a**). In addition, while all the inventories recognize high-intensity fast
landslides, the *generational-historical* and the *composite multi-temporal inventory*
records a greater extent of such failures, while a greater event number is reported by
the *composite multi-temporal* inventory (**Figure 6h**, **7i**, **8l**, and **9h**). In addition, both
the *generational-historical* and the *composite multi-temporal* inventories also report
slow landslides partially or entirely overlain by fast ones, a common geomorphological
pattern in the study area often unnoticed in the *basic-historical inventory*. This
discrepancy becomes particularly evident when comparing **Figures 6c**, **7c**, and **8c**,
where landslides-free areas in **Figure 6c** correspond to landslide bearing areas in
**Figure 7c** and **8c**, as well as to fast landslides in **Figures 6h, 7i**, and **8i**. Furthermore,
it is worth noting that rapid landslides are consistently scarce across all inventories.
Finally, as shown in **Figures 6–8**, the empty cells in the matrices represent specific
magnitude/velocity combinations that are absent from the analysed inventories. This
suggests that the morphological evolution of slopes in the study area is largely
controlled by slow-moving landslides of varying magnitudes, along with high-
magnitude fast landslides.





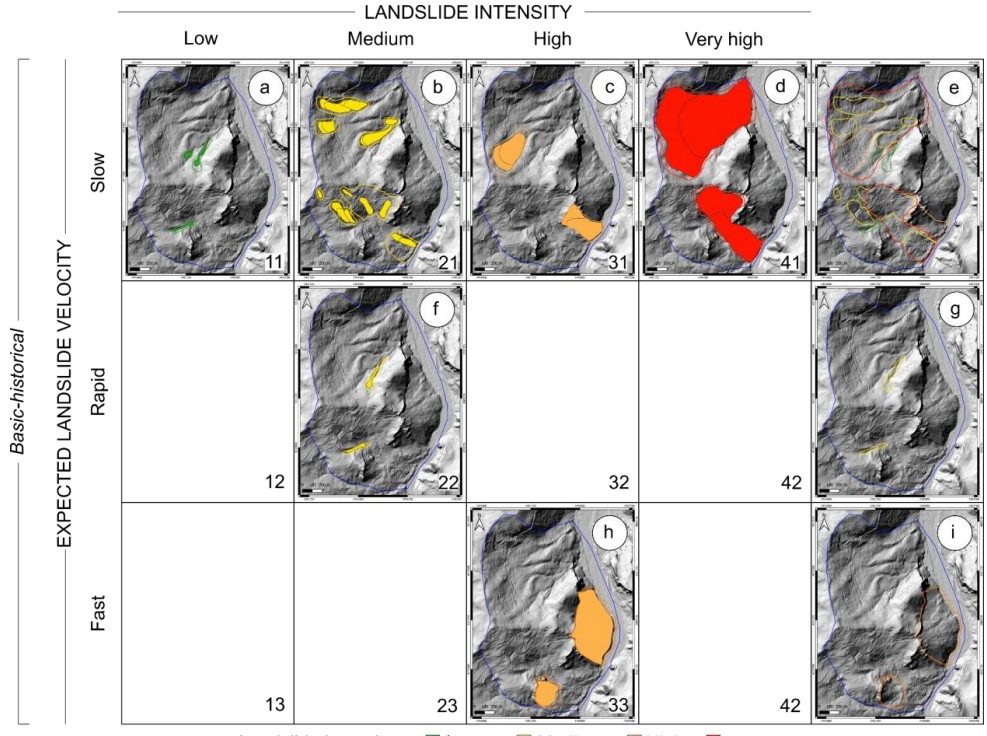

**Fig. 6** *Basic-historical* landslide inventory (**Figures 2a**, **2b**). Landslides (filled polygons) and *LHZs* (empty polygons) for slow, rapid and fast landslides classified according to their intensity (low=green, medium=yellow, high=orange, very high=red). The numbers in the bottom right corner of each map represent the landslide intensity defined according to **Section 4.2**. Base map derived from 2m LiDAR DEM (www.sitr.regione.sicilia.it).



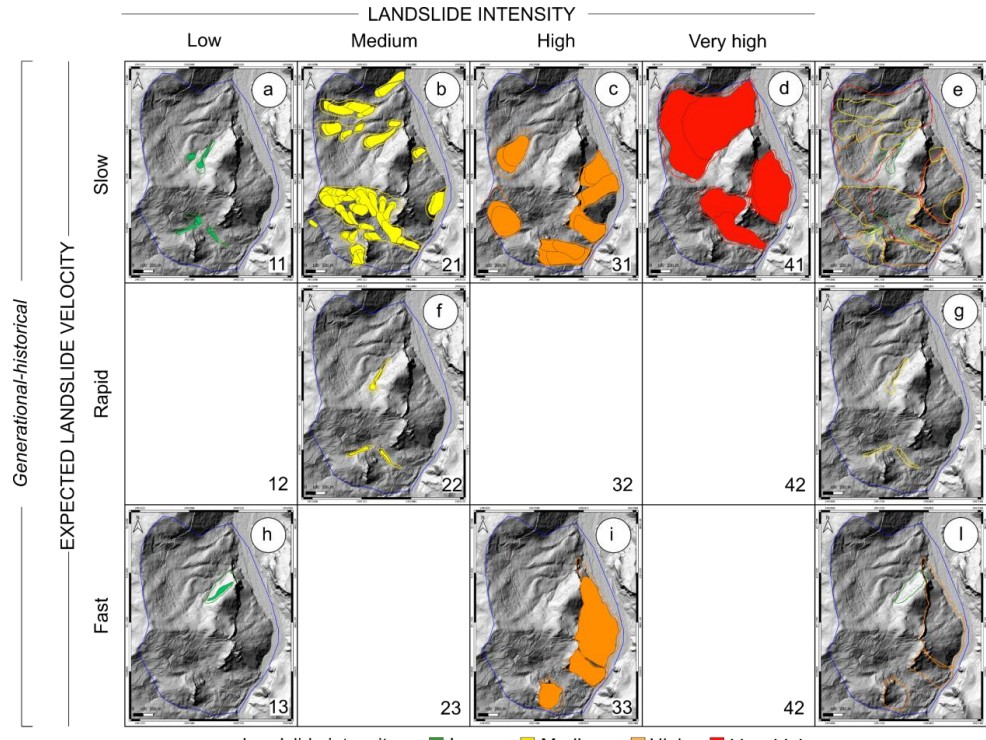

**Fig. 7** *Generational-historical* landslide inventory (**Figures 2c**, **2d**). Landslides (filled polygons) and *LHZs* (empty polygons) for low, rapid and fast landslides classified according to their intensity (low=green, medium=yellow, high=orange, very high=red). The numbers in the bottom right corner of each map represent the landslide intensity defined according to **Section 4.2**. Base map derived from 2m LiDAR DEM (www.sitr.regione.sicilia.it).

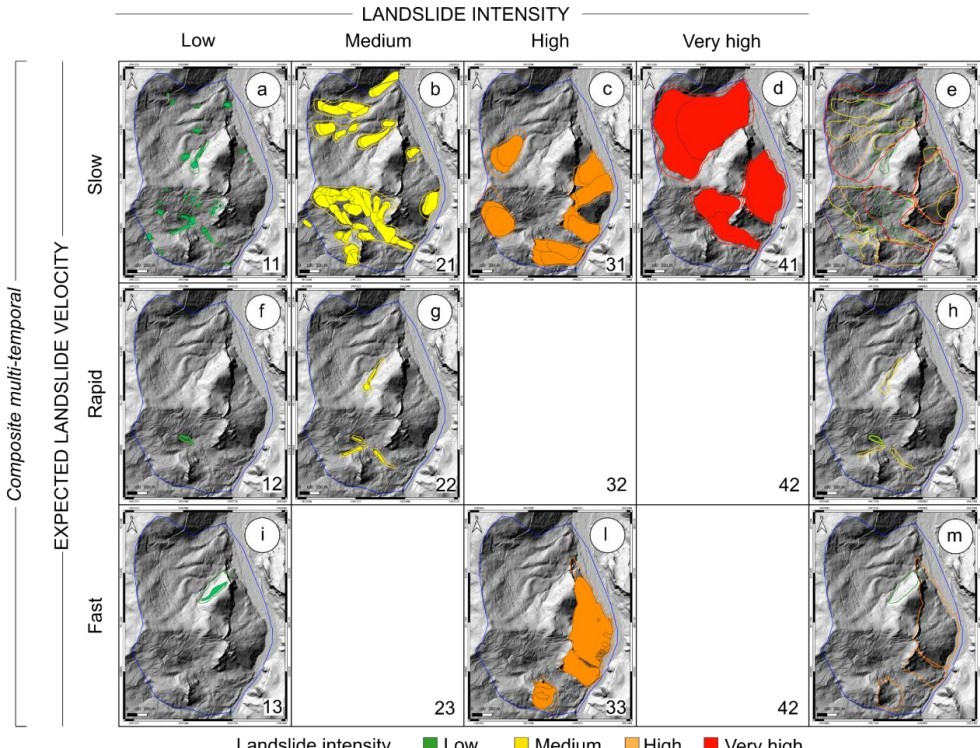

**Fig. 8** *Composite multi-temporal* landslide inventory (**Figures 2e**, **2f**). Landslides (filled polygons) and *LHZs* (empty polygons) for low, rapid and fast landslides classified according to their intensity (low=green, medium=yellow, high=orange, very high=red). The numbers in the bottom right corner of each map represent the landslide intensity defined according to **Section 4.2**. Base map derived from 2m LiDAR DEM (www.sitr.regione.sicilia.it).





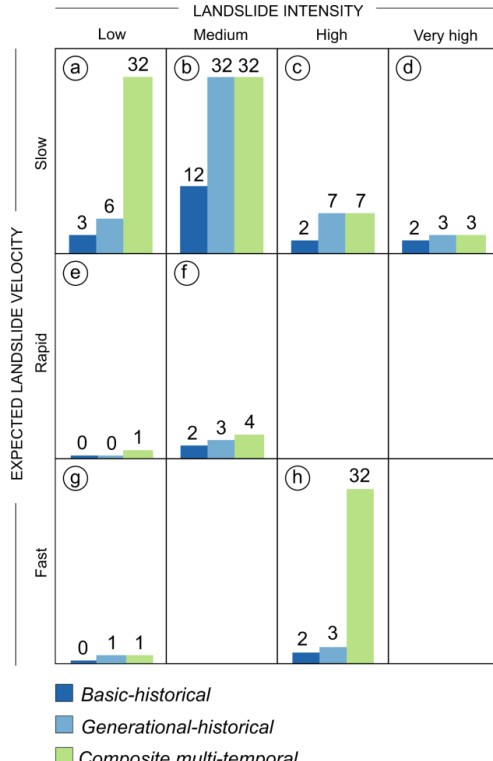

**Fig. 9** Number of landslides classified according to their intensity (low, medium, high, very high) and velocity (low, rapid, and fast) recognized considering the basic-historical, the generational-historical, and the composite multi-temporal landslide inventories (see **Figures 6-8**).

### 5.2 Landslide frequency evaluation

**Figures 10-12** show the results of the landslide frequency estimation, for slow, rapid and fast landslides, respectively. **Figure 10i-p** clearly shows that the frequency values of *LHZs* for slow landslides of all the intensity classes increase from the original (**Figures 10m-p**) to the new (**Figures 10i-l**) counting method, both applied to the *composite multi-temporal inventory,* except for the low-intensity *LHZs* (**Figure 10i, m**), which are based on the distribution of post-1955 landslides in the *composite multi-temporal* inventory (**Figure 9a**). **Figure 10** also compares the frequency values of *LHZs* derived from the *generational-historical* and the *basic-historical* inventories, as computed with the new method. The increased mapping detail increases the frequency values of landslides within most *LHZs* as well as their spatial coverage.

**Figure 11** shows that, in our case study, switching between inventories frequency counting methods does not impact the (negligible) role of rapid landslides in the hazard assessment.





Finally, **Figure 12** shows that the new method considers the highest frequency (4, very
high) for fast landslides *LHZs* in the southern part of the study area, as opposed to the
original method which counts a lower frequency (3, high), as derived from the
multitemporal inventory map. This is consistent with the specific condition posed by
the presence of rockfall areas, which are assumed to be subject to the highest
frequency value in case of unavailability of multi-temporal information (i.e. having only
a historical inventory map).

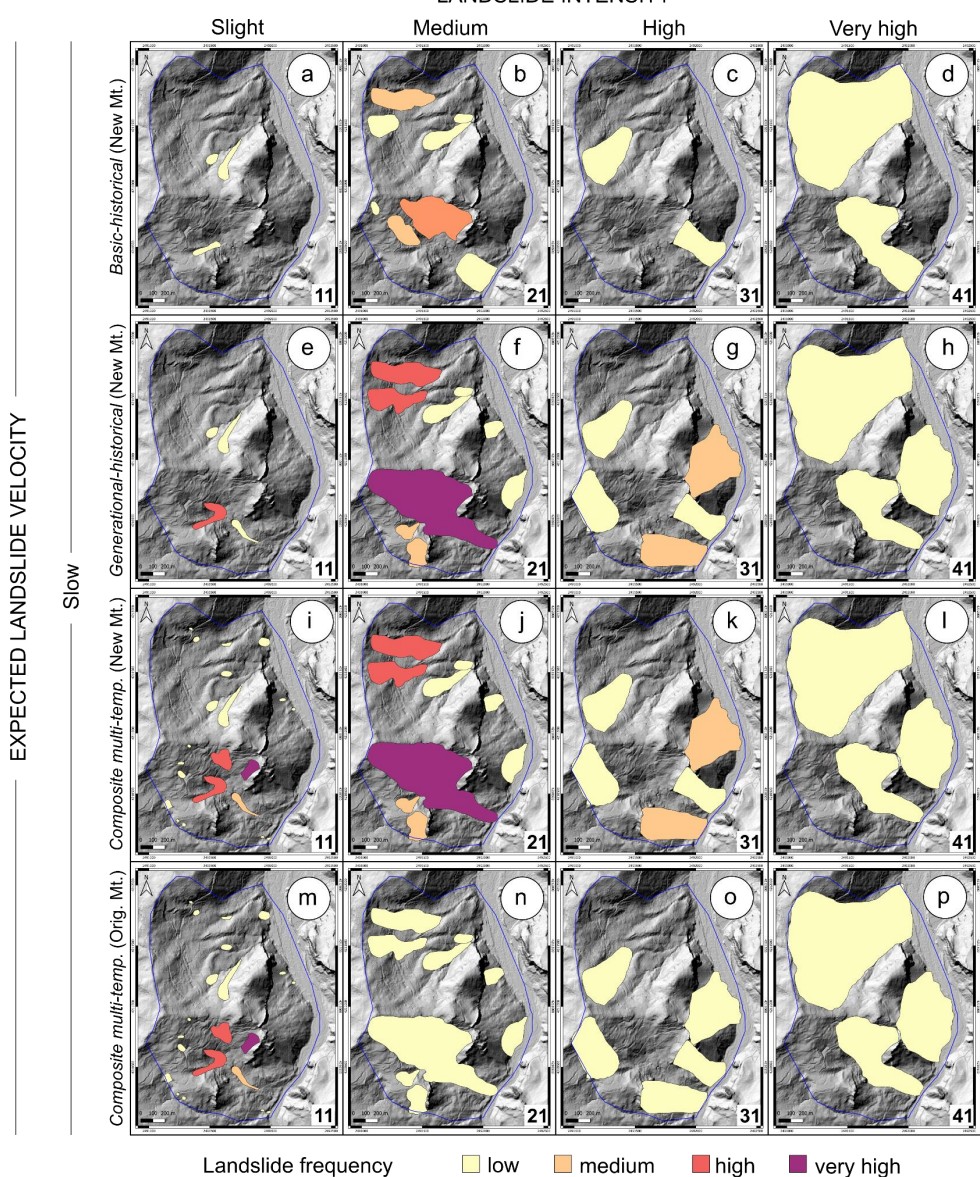

**Fig. 10** Landslide frequency for slow landslides *LHZs* estimated considering (i) the original
method (m, n, o, p), and (ii) the new method applied to the *composite multi-temporal* (i, j, k, l),



*generational-historical* (e, f, g, h) and *basic-historical* inventories (a, b, c, d). The numbers in
the bottom right corner of each map represent the landslide intensity defined according to
**Section 4.2**. Base map derived from 2m LiDAR DEM (www.sitr.regione.sicilia.it).

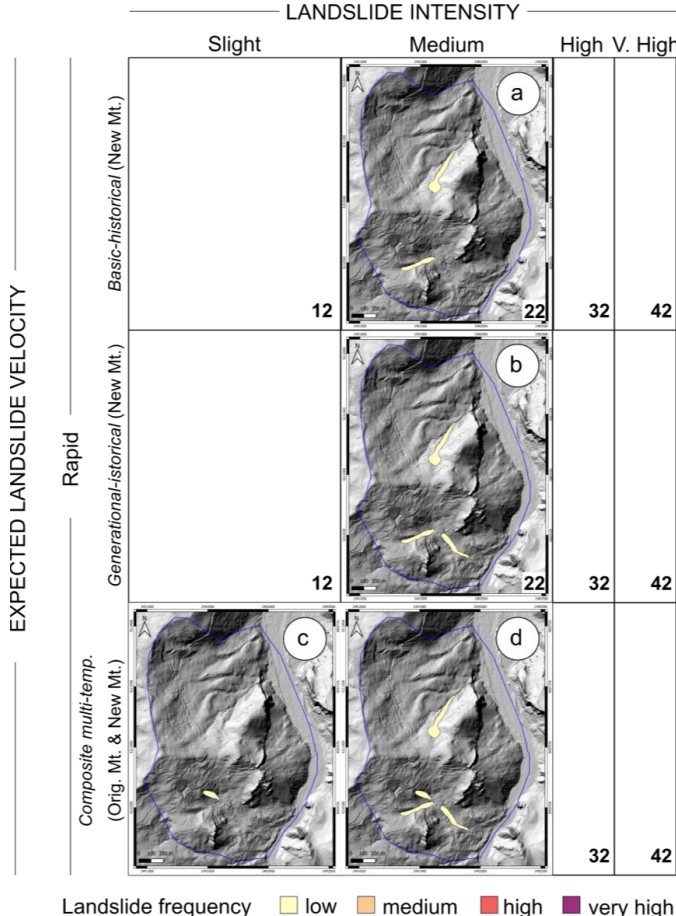

**Fig. 11** Landslide frequency for rapid landslides *LHZ*s estimated considering (i) the original
method applied to the *composite multi-temporal* inventory (c, d), and (ii) the new method,
applied to the *generational-historical* (b) and *basic-historical* (a) inventories. The numbers in
the bottom right corner of each map represent the landslide intensity defined according to
**Section 4.2**. Base map derived from 2m LiDAR DEM (www.sitr.regione.sicilia.it).

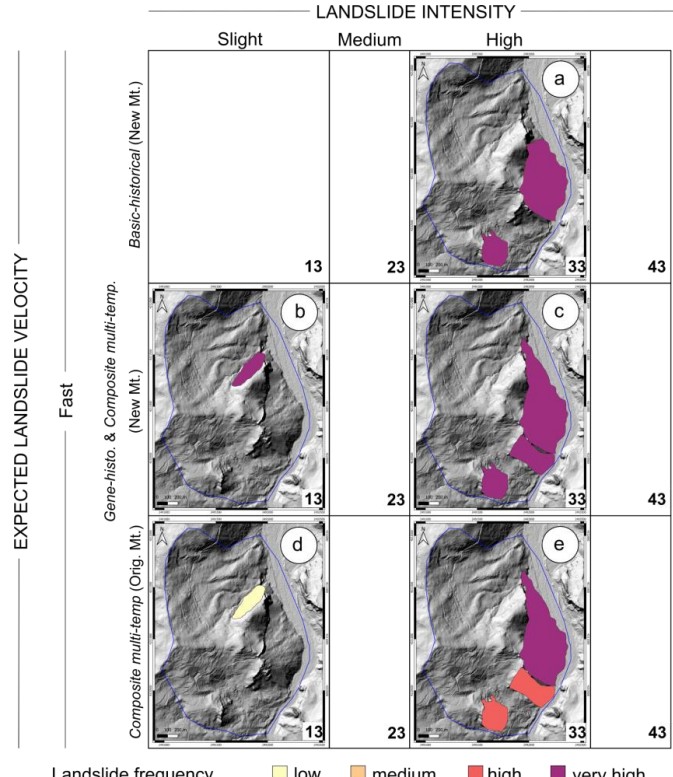

Fig. 12 Landslide frequency for fast landslides *LHZs* estimated considering (i) the original method (applied to the *composite multi-temporal* inventory, bottom row), and (ii) the new method, applied to the *generational-historical* and *basic-historical* inventories. The numbers in the bottom right corner of each map represent the landslide intensity defined according to **Section 4.2**. Base map derived from 2m LiDAR DEM (www.sitr.regione.sicilia.it).

### 5.3 Ground motion of landslide polygons in potential reflective areas

The potential reflective area, indicated by the green polygon in **Figure 13a**, is derived by the land cover/use map released by Regione Siciliana (see **Section 3.4**), considering and merging the following classes: (i) residential areas, villages and buildings; (ii) roads and road infrastructure; (iii) arid grasslands, junipers, garrigue; (iv) low shrubland with cistus, rosemary, and mediterranean sclerophyllous plants; (v) bare rocks, cliffs, streams and alluvial beds.

Out of the 4560 PS from ERS, ENVISAT and COSMO-SkyMed in the study area, 98% fall within the potential reflective area, accounting for 52% of the study area.

In **Figure 13a** the ground motion signal is represented according to the absolute value of the velocity of each PS (see **Section 4.5**). Inspection of the figure shows: (i) a predominantly stable area corresponding to the village of Militello Rosmarino (See "anthropic surfaces" in **Fig 1c**), where most PS exhibit velocity ($V_{PS}$) values below 1 mm/yr; (ii) a moderately unstable area ($1 < V_{PS} < 5$ mm/yr) along the eastern boundary



of the study area; and (iii) a highly unstable area ($V_{PS}$>12 mm/yr) in the western and
southern sectors of the study area, where part of the built-up area is developed.
In **Figure 13b**, PS data weighted by the velocity attribute field are displayed as a
density map, computed using a 30 m radius moving window, and are overlaid onto the
*composite multi-temporal inventory*, classified by landslide age. The figure shows that
the highest PS density clusters are located within pre-1955 landslides, predominantly
of the second and third generation, in the southwestern part of the study area.
Conversely, PS density clusters show poor spatial correlation with post-1955
landslides.

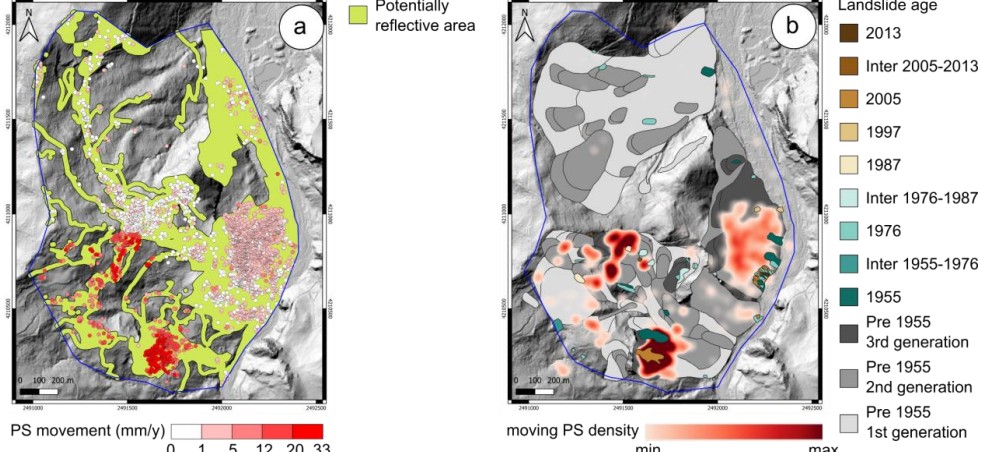

**Fig. 13** (a) PS movement (www.doris-project.eu) overlapped to a potentially reflective area
elaborated from the 1:10,000 scale land cover/use map (www.sitr.regione.sicilia.it), (b) PS
density overlapped to the *composite multi-temporal* landslide inventory map. Base map
derived from 2m LiDAR DEM (www.sitr.regione.sicilia.it).

## 6. Discussion

The main novelty of the workflow to assess landslide hazard proposed in this paper
(see **Figure 3**) is the use of a *generational-historical* landslide inventory as base map,
instead of a multi-temporal one as requested by the original method of *Cardinali et al.*
*(2002)*. The rationale supporting this proposal derives from the widely verified
observation that landslides occur where they have already occurred in the past (i.e.,
legacy effect, see e.g. *Samia et al., 2017a*; *2017b*) and that evidence of landslides
recurrence seems to be already embryonic in a historical inventory map drawn with
generational criteria. What is missing in a historical inventory is a clearly defined time
window for observing landslide recurrence.
Looking at the case study presented in this paper and shown in **Figure 2**, we only can
affirm that: (i) all the landslides occurred before 1955, (ii) they are all recognizable
without significant modification in the 2005 air photos (see **Section 5.1**), (iii) and they
evolved through subsequent generations, partially or totally overlapping to the first





failures. Therefore, considering that there are no noticeable changes in the
appearance of the pre-1955 landslides between the photos of 1954 and 2005, the
reasonable assumption we make in our procedure is to consider also valid today and
for the next future the evolutionary trend recognized for the pre-1954 landslides
through generational mapping.
In the case study, results suggest that, except for small and shallow landslides, multi-
temporal mapping is not decisive for the definition of the hazard, and may therefore
be skipped, with positive consequences on the efficiency of the landslide hazard
estimation. Multi-temporal inventories are more demanding than historical landslide
inventory maps (*Guzzetti et al., 2012*), the last one requiring less data (i.e. aerial
photographs) and time. In addition, compiling a multi-temporal landslide inventory
does not ensure high mapping detail and consequent data reliability, because it
focuses only on few small recent landslides while the older landslides remain
unchecked in the background and are often heterogeneously represented, affected by
mapping subjectivity, dependence on the acquisition method, and incompleteness.
Therefore, before proceeding to hazard estimation, a preliminary critical analysis of
the available historical inventories is needed.
**6.1 Evaluation of landslide inventory maps**
Our proposed method for evaluating the historical landslide inventories is
straightforward and relies on systematically collecting metadata (**Table 2**) and
qualitatively comparing it against reference inventories. This approach ensures that
inventories are not used without a clear understanding of their underlying data
quality.The qualitative comparison revealed that the *generational-historical inventory*
aligns well with reference inventories, whereas the *basic-historical inventory* exhibits
significant discrepancies. Such differences are primarily attributable to the level of
mapping detail: the *basic-historical inventory*, designed for broader regional
assessments under resource constraints, inherently simplifies complex landslide
geometries. This simplification results in fewer detected landslide generations, lower
overall density of the landslide spatial distribution, and a larger value of the smallest
mapped feature (i.e. an indicator of incompleteness, *Guzzetti et al., 2002, Malamud et*
*al., 2004*). In contrast, the *generational-historical inventory* benefits from high-
resolution LiDAR data and photo-interpretation using advanced digital stereoscopes,
which enhance the detection of landslides obscured by vegetation or subsequent
slope failures.
Such differences in mapping detail have direct implications for landslide hazard
evaluation. A non-generational historical inventory, while capable of supporting a
rough hazard assessment, is inherently limited by data gaps that can lead to
underestimations of both landslide frequency and the extent of hazard zones. Thus,
methodological rigour and higher mapping resolution are critical for accurate hazard
analysis.





In summary, our findings underscore the importance of detailed and systematic
inventory compilation. While basic inventories may provide enough information for
regional knowledge, a generational approach is essential for reliable hazard
evaluation, ultimately providing a more accurate basis for decision-making.
**6.2 Landslide hazard evaluation by using historical and multi-temporal inventory**
**maps**

Unlike historical inventories, given a decadal revisit time of aerial photographs, multi-
temporal inventories record even small (~$10^2$ m$^3$) volumes of material mobilized, since
most of their traces remain discernible in the available images (e.g., scarps, trenches,
or disruptions in land-use patterns, e.g., *Galli et al 2008, Ardizzone et al., 2024, Bucci*
*et al., 2021, Zumpano et al., 2019*).
Our mapping data provides further evidence to the increasing Literature (*Samia et al.,*
*2017b, Temme et al., 2020, Chen et al., 2024*) that the signal of recent evolution
captured by the multi-temporal inventory tends to cluster around or inside areas of pre-
existing instability as defined by the generational-historical inventory, and, less
prominently, by the non-generational historical inventory. This observation supports
our proposed frequency-count adjustment, which refines the hazard estimation
procedure by incorporating the number of landslides recorded prior to 1955 in
historical inventories.
**Figure 14** compares the landslide frequencies estimated for slow *LHZs*, the prevalent
zones in our study area (see **Figure 10**), applying both the original and the revised
methods to the three available inventories.
The first two columns of **Figure 14** demonstrate that the revised method consistently
yields more conservative estimates compared to the original method, formalizing
higher frequency classes.
For the landslides not captured by the multi-temporal inventory (i.e., those
documented solely in the historical one), the original hazard assessment method
assigns a default frequency of 1. As a result, the hazard of these landslides is
determined solely by their magnitude, and in particular by their size, since more than
80% pertains to the same typology (slide and slide-flow) and expected velocity class
(slow moving). Consequently, according to the original method, the most dangerous
landslides are necessarily the smallest of the multi-temporal inventory and the largest
of the historical inventory, while the hazard of all medium-size landslides
systematically results underestimated. This is a first order problem of the original
method, since the mapping itself seems to indicate quite the opposite, suggesting
instead that the medium-size landslides are the more dynamic, developing through
successive generations in portions or at the margin of larger previous landslides.
Furthermore, the composite multi-temporal inventory (**Figure 14g**) reveals that while
the inclusion of multi-temporal landslides enriches the data for smaller, scattered
*LHZs*, it does not substantially influence the overall hazard characterization, which
remains predominantly governed by landslides portrayed in the generational-historical



inventory (**Figure 14d**). Given that landslide hazard is a function of both intensity and
frequency, our revised frequency-counting method, applicable across all magnitudes,
significantly impacts the hazard assessment, as reflected in the hazard matrix
presented in **Table 2** (**Section 4.4**).

**6.3 Comparison of PS dynamics and frequency of slow landslides within the**
**_LHZs_**

For the _LHZ_s of slow-moving landslides - which are the 80% of the pre-1955 landslides
(i.e. slide and slide-flow types) - the frequency value obtained with the two methods
described before can be compared with the PS velocity data present within each _LHZ_.
Going more in detail, the PS within each _LHZ_ for slow landslides were selected,
isolated from the others, and analyzed by a contouring of their velocity data. The
objective of the analysis is to highlight the velocity gradient within each _LHZ_ for slow
landslides, considering it a proxy of the evolutionary trend - and therefore of the
frequency - of the landslides within each _LHZs_. The third column of **Figure 14** shows
the contours of the PS velocity computed within each slow _LHZs_ from the _basic-_
_historical_ (**Figure 141c**), _generational-historical_ (**Figure 14f**) and _composite multi-_
_temporal_ (**Figure 14i**) landslide inventory map. The PS falling outside the _LHZs_ were
also reported in **Figure 14** and represented as points classified according to their
absolute velocity values. In **Figure 14f** and **Figure 14i**, almost all the PS falling outside
the _LHZs_ are stable and clustered roughly at the center of the study area, where the
historical center of Militello Rosmarino is located. In contrast, in **Figure 14c**, many
more PSs fall outside the _LHZs_, clustering locally in large areas affected by coherent
and significant deformation rates. Comparing the three figures, it can be clearly
observed that the inadequate _LHZs_ coverage in **Figure 14c** is the consequence of the
unrecognized landslides in the _basic-historical_ inventory, thus demonstrating both the
effective existence of these landslides and their state of activity. Overall, PS data
indicate the presence of two areas with low and high movement, respectively in the
South and East sectors of the study area, already shown in **Figure 13** in
correspondence of the more dense pre-1955 landslides clusters. A visual inspection
of **Figure 14d**, **14e**, and **14f** shows that these areas with low and high movement are
those characterized by medium and very high landslide frequency recognized by the
new method, while they were not recognized by the original method, which only
considers the frequency related to small, superficial and recent (after 1955) landslides.
The combined inspection of **Figure 13** and **Figure 14** reveals that the areas of dense
clusters of overlapping landslides (i.e. landslide generations subsequent to the first)
are also the areas where PS scattering indicates greater deformations over a time
span of two decades, covering by ERS (1992-2001), ENVISAT (2003-2010) and
COSMO Sky-Med (2011-2012) PS data. The trend is confirmed by PS data provided
by the European Ground Motion Services (https://egms.land.copernicus.eu/) for the
period 2018-2022. In other words, the zones (i.e. _LHZs_) subject to the evolution of
multi-generational landslides in the past are also generally characterized by higher





recent instability. We interpret these pieces of evidence as the result of local
morphological and hydrological perturbations induced by the occurrence of the first
failure which promote the evolution of landslides of subsequent generations in
materials with residual geotechnical properties, and, more in general, determinate the
maintenance of conditions of general instability. This would explain why the frequency
calculated with the new method better matches the clustering of the unstable PS.
Anyway, as shown in **Figure 13**, PS data cannot be used everywhere, especially in
not urbanized areas, where their coverage remains poor. For this reason, PS data
cannot be included within our procedure, which by definition aims to be applied to an
entire territory covered by a landslide inventory map, independently from the land use
and coverage. On the other hand, the use of PS data becomes significant after the
application of the procedure as an independent measure of the recent activity of slow-
moving landslides, which can be compared with the frequency (i.e. past activity) of the
related *LHZs*. Also, more importantly, since landslide hazard is expressed using a
multiple digit index that portrays all the variables used, PS data can be used as
additional synthetic information to be added at the end of the hazard assessment
process. As an example, in **Figure 14** we visualize PS data together with *LHZs* of slow
landslides. In the figures we used three classes of *LHZ* thickness to highlight: (i) the
absence of PS or the presence of stable PS with velocity close to 0 mm/y, (ii) the
presence of moderately unstable PS with velocity lower than 8 mm/y, (iii) the presence
of highly unstable PS with velocity higher than 8 mm/y. Apart from the first row of
**Figure 14**, which suffers from the incompleteness of the *basic-historical* inventory, the
second and third rows of **Figure 14** indicate a substantial matching between the *LHZs*
with non-negligible PS derived deformations and *LHZs* with frequency greater than 1.
The evidence depicts a strong linkage between the long term (i.e. centuries) and short
term (i.e. years) evolution of slow-moving landslides, with substantial implication for
their hazard. Overall, we consider the additional information provided by PS data,
together with the hazard zoning output, a major advantage of the presented workflow,
giving decision makers great flexibility in deciding which area exhibits the highest
hazard, also in the light of the variability of the available ground motion values.

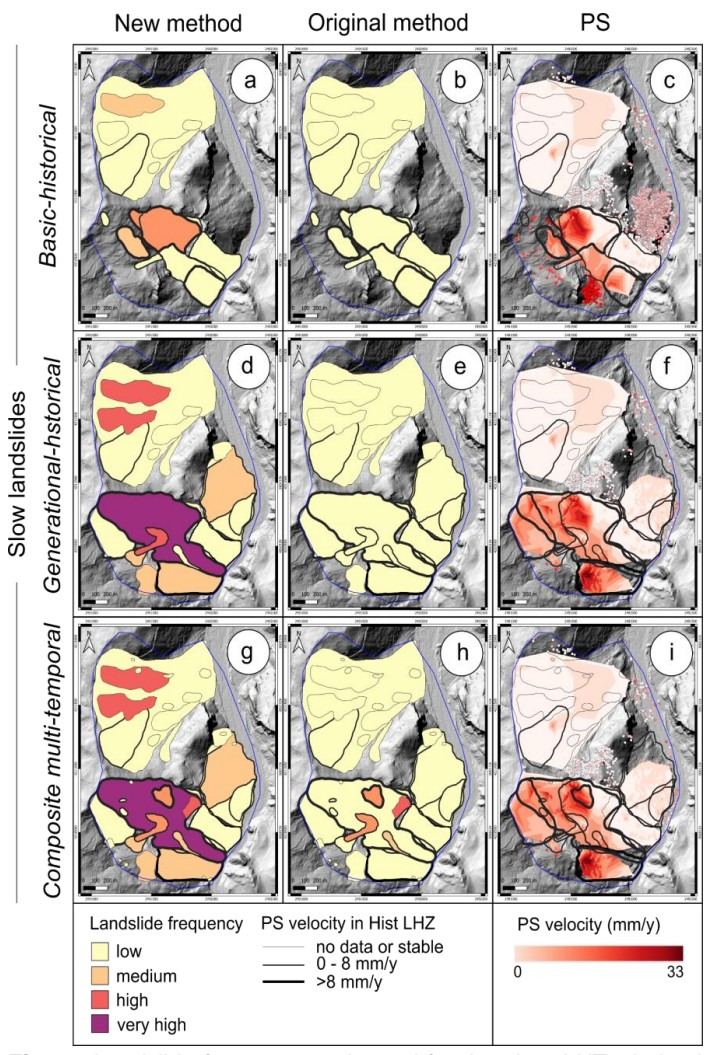

**Fig. 14** Landslide frequency estimated for the slow *LHZs* derived from the *composite multi-temporal* (g, h), *generational-historical* (d, e), and *basic-historical* (a, b) inventories considering the new (a, d, g) and the original (b, e, h) methods. **Figures c**, **f**, **i** show contour of the PS movement overlapped to the slow *LHZs* from the considered inventories. Base map derived from 2m LiDAR DEM (www.sitr.regione.sicilia.it).



## 7. Conclusion

Landslide hazard assessment is a complex task which is based on the elaboration of landslide inventory maps and contains three components: a spatial one (where landslides might occur), an intensity one (how large or destructive landslides might be) and a temporal one (when or how often landslides might occur). In this paper we investigated how different landslide inventory maps, in particular a *basic-historical*, a *generational-historical*, and a *composite multi-temporal*, may influence the landslide hazard assessment. We proposed a revised version of the methodology described in *Cardinali et al. (2002)*, in this paper named as original method, consisting in the use of historical landslide inventory maps, instead of multi-temporal landslide inventory maps, which require longer times and much more data (e.g. multiple flights), and therefore cannot be realized over a large area. The main difference between the original method and the new method shown in this paper, consists in a different counting of landslide frequency. In particular, in the original method the landslide frequency was determined considering only the information derived from the multi-temporal landslide inventory map, counting the number of recent landslides observed within a *LHZ* (i.e. the area of possible, or probable, short-term evolution of an existing landslide, or a group of landslides, with similar characteristics). In this work, the new method extends the count of the number of events also to landslides that occurred before 1955, during an undefined time period, considering the generational classification approach of the landslides. A further difference between the original method and the new method, consists in the comparison of PS dynamics with the frequency of slow landslides within their *LHZs*, which was not considered in the original procedure.

The workflow was applied in an area surrounding the village of Militello Rosmarino (NE Sicily, Southern Italy) that is prone to landslides of different types and sizes. Before applying the procedure for hazard estimation, we set a method to evaluate the examined inventories in order to consider a qualitative measure of the uncertainty derived by the landslide data itself.

Overall, the results show that the widest recognized landslides are those older than 1955, reported in the *generational-historical* inventory and mainly classified as slides (i.e. slow landslides), which represent about 97% of the total landslide area. Instead, the landslides recognized after 1955, reported in the multi-temporal inventory, represent only 3% of the total landslide area and almost entirely fall into areas already characterized by pre-existing mass movements.

The comparison of different inventories indicates that the contribution of multi-temporal landslides (i.e. the landslides recognized after 1955) only enriches the information relating to the smaller and more scattered *LHZ*s. Instead, multi-temporal landslides do not contribute to characterizing the landslide hazard of the study area, which instead is predominantly controlled by previous landslides, already present in the *generational-historical* inventory, and only partially, in the *basic-historical* inventory.

The comparison of different methods of landslide frequency estimation indicates that the new method always leads to the formalization of higher frequency classes



compared to the original method, especially for slow landslides of medium and high
magnitude that typically evolve through multiple generations.
The main conclusion that can be drawn from our results is that, except for small and
superficial landslides, multi-temporal mapping is not decisive for the definition of the
hazard. It is instead important to base landslide hazard analysis on a *generational-*
*historical* inventory that adequately characterizes the complexity of landslide clusters.
On the other hand, caution must be posed in using a *basic-historical* inventory, which
potentially may contain partial or too generalized landslide information.
The application of the PS techniques confirms that the areas more unstable are those
with the higher density of pre-1955 landslides of different generations recognized in
the *generational-historical* inventory, which translates into the higher values of
frequency and consequently of hazard. This further finding allows us to suggest the
use of PS data as additional synthetic information to be added at the end of the hazard
assessment process, benefiting the territorial planning choices, which - we expect -
could base on the definition of hazard classes starting from the hazard indexes
obtained from the workflow here presented.
Overall, our procedure puts landslide mapping back at the center of the concept of
hazard, establishing and verifying an adequate data acquisition method for its
definition. Since there is nothing peculiar or specific in our case study, applying the
method here presented in other morphological contexts, even spanning over much
larger study areas, we expect an improvement in landslide hazard estimation similar
to those illustrated by our results. We therefore encourage the application of our
procedure in other environments and with other inventories, and the comparison with
results from other data-driven hazard assessment methods, to shed light on future
research needs in this field.

## Author contribution

**Marco Donnini**: Conceptualization, Data curation, Investigation, Methodology, Writing
(original draft preparation), Writing (review and editing). **Francesco Bucci**: Conceptualization,
Data curation, Investigation, Methodology, Writing (original draft preparation), Writing (review
and editing). **Michele Santangelo**: Conceptualization, Investigation, Methodology, Writing
(review and editing). **Mauro Cardinali**: Investigation, Methodology, Resources, Writing
(review and editing). **Paola Reichenbach**: Methodology, Project administration, Supervision,
Writing (review and editing)

## Credits and Acknowledgments

The research was developed in the framework of the "*DORIS (Ground Deformation Risk*
*Scenarios: an advanced Assessment Service)*" project founded by the European Union
Seventh Framework Program (FP7/2007–2013) under grant agreement No. 242212, and of
the "*PON Governance e capacità istituzionale 2014–2020*" project, contract CIG 6983365719.



**Ancillary materials**

### A.1 Typological classification of landslides

To define the landslide typologies, as shown in *Ardizzone et al. (2023)* and in *Guzzetti et al. (2012)*, landslides can be defined according to the classifications of *Cruden and Varnes (1996)*, and *Hungr et al. (2014)*, following the schema shown in **Table A1**.

**Table A1** Description of landslide type according to *Varnes (1978)*, *Cruden and Varnes (1996)*, and *WP/WLI (1990, 1994, 1995)*.

| Type | Sygn | Description |
|---|---|---|
| Slide | *s* | Slides are movements that create a general concavity and convexity on the topographic surface without significant de-articulation. Surface ranges from a few dozen square meters to a few square kilometers. |
| Earth Flow | *f* | Earth flows are landslides characterized by the movement of material, usually clayey down a gentle slope in the form of a fluid. Flows often have a distinctive, upside-down funnel shaped deposit where the landslide material has stopped moving. The earth flows are mainly distributed within other pre-existing landslides. |
| Debris Flow | *df* | Debris flows are frequent where debris production is abundant (fractured areas, landslide deposits, talus). They have narrow and elongated shapes characterized by: (i) a source area, (ii) a generally narrow and elongated channel and (iii) an accumulation area that at the foot takes on the characteristic convex shape. Surface ranges from a few dozen square meters to a few square kilometers. |
| Rockfall | *rf* | Falls are landslides that involve the collapse of material from a steep slope or a cliff. A fall-type landslide results in the collection of rock or debris near the base of a slope. |
| Rockfall area | *rfa* | Rock fall area is an area characterized by widespread rock fall phenomena, where single rock fall is difficult to recognize. |
| Slide-Flow | *sef* | Slide-Flows are a complex or composite landslide type. In general they are characterized by the presence of two of the types of movement described above. Slide-Flows may have occurred at different times (complex) or simultaneously in the same area (composite). |





**A.2 Definition and delineation of the Landslide Hazard Zones (*LHZs*)**

The areas of evolution of existing (mapped) landslides are named by the authors Landslide Hazard Zones (*LHZs*), and are defined as areas of possible (or probable) short-term evolution of existing landslides with similar characteristics (i.e. of type, volume, depth, and velocity). A *LHZ* is therefore a "landslide scenario" delimited using geomorphological criteria considering (i) the partial or total reactivation of existing landslides, (ii) the lateral, head (retrogressive) or toe (progressive) expansion of the existing landslides, and (iii) the possible occurrence of new landslides of similar type and intensity. Different *LHZs* can be determined for each type of failure observed on an elementary slope (e.g. fast-moving rock falls, rapid-moving debris flows, slow-moving earth-flow slumps or compound failures).

**A.3 Estimation of landslide volume and velocity**

As observed by *Cardinali et al. (2002),* unlike natural hazards such as earthquakes or volcanic eruptions, a universally acknowledged measure of landslide intensity is not recognized in the literature. Following *Hungr (1997)*, landslide intensity can be considered as a function of landslide volume and expected velocity, proxies of the landslide destructiveness. Landslide volume can be estimated starting from the landslide area and on the basis of landslide type, using the **Eq. [1]** of *Guzzetti et al. (2009)* for slides and slide-flows, and the **Eq. [2]** of *Innes (1983)* for flows and debris flows, as shown in **Table A2**.

$$V_{slide} = 0.074 \times A_{slide}^{1.45}$$ **Eq. [1]**

$$V_{flow} = 0.0329 \times A_{flow}^{1.3852}$$ **Eq. [2]**

In the equations, $V_{slide}$ and $A_{slide}$ represent, respectively the landslide volume and area of slides and slide-flows, while $V_{flow}$ and $A_{flow}$ represent, respectively the landslide volume and area of flows and debris flows. For rockfall (*rf*), as well as for rockfall areas (*rfa*) there are no empirical relationships relating areas to volumes. In these cases, a reliable measure of the magnitude is the volume of the maximum expected boulder involved in *rf* and/or recognized within *rfa*, which can be estimated through images interpretation and/or field survey.

**Table A2**: Schema for estimating landslide volume.

| Landslide typology | Landslide velocity | Landslide Volume |
|---|---|---|
| Slide (*s*) | Slow | Eq. [1] |
| Flow (*f*) | Slow/Rapid | Eq. [2] |
| Debris flow (*df*) | Rapid | Eq. [2] |
| Rockfall (*rf*) | Fast | Largest boulder volume |



| Rockfall area (*rfa*) | Fast | Largest boulder volume |
| Slide-flow (*sef*) | Slow | Eq. [1] |

856

According to *Cardinali et al. (2002)*, the expected landslide velocity can be discretized
into three classes (1: slow, 2: rapid, 3: fast) following the schema shown in **Table A2**
where slow landslides are slides and slide-flows; rapid landslides are debris flows; fast
landslides are rockfalls and rockfall areas; while flows can be considered slow or rapid
(see e.g. *Cruden and Varnes, 1996*).



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
