# Peer review of "The effect of different landslide mapping approaches on the geomorphological assessment of landslide hazard"

_EGUsphere, 2025_

## Author Comment (AC1)

*Responses to Reviewer #1*

I started to read the manuscript with great interest as in the abstract the paper claims to create a new method to assess landslide hazard and puts inventory at the center of landslide hazard assessment. However, I am not satisfy those claims are substantially supported by the manuscript and my major observations are as follows:

*We thank the reviewer for his critical reading of the manuscript and for his detailed comments. We are pleased to hear that the abstract has attracted great interest. We will improve the paper to ensure this interest remains high throughout the manuscript.*

1. The literature review/introduction part of the manuscript is very weak as it does not properfly define the rational behind this research and does not explain the research gaps (why this work needs to be done) and what are the existing gaps in the scientific literature. For example, recent works on physically based analysis for small regions and statistical/ML approaches for large regions has significantly well defined the landslide hazard in terms of exceedence probability, susceptibility and hazard itself. Gap on those works must be stated and your work must be well posed to solve those gaps. (e.g. why your work is needed when the work of Guzzetti 2005 on probabilistic approach or recent improvements on that can define landslide hazard).

*1. The scientific advances in landslide hazard estimation cited by the reviewer concern physically-based analysis and statistical/ML approaches. Our work, however, uses a semi-quantitative heuristic approach, which, unlike modeling approaches, is little explored in the scientific literature. The method we propose does not express landslide probability in modeling terms (probabilistic or deterministic), but defines landslide hazard areas based on landslides that have already occurred and the potential evolution zones for individual landslides or groups of landslides. Within these areas, it defines the hazard level based on the magnitude, velocity and frequency of landslides that have already occurred and are expected. While highlighting certain limitations, we maintain that our method represents an important advance in the field of semi-quantitative landslide hazard estimation because: i) it utilizes and optimizes information on landslide intensity and recurrence contained in inventory maps; ii) it takes into account the spatial and temporal dimensions of hazard; iii) it establishes a logical and reproducible procedure; iv) it overcomes the subjectivity of traditional geomorphological approaches; v) it provides maps directly applicable to land-use planning. We will introduce and appropriately argue these points in the new version of the manuscript. However, the presentation of a new method for landslide hazard assessments is not the only result of this work. Indeed, as the title suggests, the main objective of this paper is to demonstrate and measure the effect of different mapping approaches on landslide hazard estimation. In the new version of the manuscript, we will better highlight this additional result and its scientific relevance.*

2. The method defined for this work needs to be clarified as method section ends abruptly at the section comparing with PS points. I did not understand what is the end goal of your landslide hazard assessment. does it provide landslide hazard zonation (like class low medium high) or does it provide quantitative measure. From the introduction section my impression is that you have semi-quantitative approach for defining hazards.

*2. The reviewer is right. In the new version of the manuscript we will define the method more explicitly in all its steps so that the end goal of our landslide hazard assessment is clear. This will also require adding new text and a new figure in the results section.*

3. Can we define landslide hazard just based on the frequency of past landsldies? Major issue that I have with the approach is that you assume the landslide will occure in the same zone if it

has occured in the past observation and vice versa. I would asuume the landslides often change the terrain characterstics changing the landslide dynamics and frequency, thus changing the hazard.

*3. Not exactly. Our procedure for defining landslide hazard takes into account the magnitude, expected velocity, spatial distribution, and finally, the frequency of past landslides. The underlying concept of our method is: learning from the evolution of past landslides to estimate the characteristics (and therefore the hazard) of future landslides. This concept is based on the widespread evidence that past landslides worsen the geotechnical conditions of the affected terrains and disturb the morphological stability and geo-hydrological circulation of the affected slopes, thus generating and maintaining conditions of instability in the evolution area of the considered landslides, until a new equilibrium is reached. The information contained in the geomorphological landslide inventory maps we use in this work covers an indefinite time window, but realistically "photographs" the landslide trend — still evolving — over the order of (at least) hundreds of years. It can therefore be assumed that similar conditions of instability will persist in the future for a similar period.*

4. How is the landslide volume/kinetic energy estimated? is it through sterio-images or based on the area of landslide polygons. Since area can be a proxy of volume why not use area itself, which can be measured with higher precision and accuracy?

*4. In the main text it is perhaps not clear enough that this information is reported in the ancillary materials at the end of the manuscript. In the new version of the MS we will indicate more clearly that this part is specified in the ancillary materials. However, in summary, we estimate the intensity of a landslide as the combination of expected velocity and magnitude. As a proxy for velocity, we use the typological classification of landslides derived from photointerpretation analysis, establishing classes of landslides as slow (slides and complex landslides), rapid (earthflows and debris flows), and fast (rockfalls). As a proxy for magnitude, we use volume (derived from the area through empirical relations). We consider volume to be the most appropriate measure of magnitude because it takes into account both landslide area and depth.*

5. Your workflow primarily maps frequency and then treats it as hazard for planning ("frequency counting", "landslide frequency zonation" and "LHZs" are the operative layers). That's consistent with the classic geomorphological tradition, but it does not explicitly model when landslides are likely to recur. Therefore, it looks more like landslide inventory analysis to understand their spatial frequency.

*5. It is important to point out that the hazard determined by our procedure is based not only on spatial frequency, but also on the magnitude and expected velocity of landslides. However, the reviewer's observation is correct: our method does not explicitly model when landslides are likely to recur. This is because the spatial frequency we determine derives from the analysis of the geomorphological inventory, which (as mentioned in point 3) does not cover a defined time window.*

*Actually, in this paper, we also apply the procedure to a multitemporal inventory, which contains landslides that occurred from 1955 to 2013. In this case, the time window is defined and a frequency rate can be established. However, in this time frame, we observe only shallow, small-scale landslides, and consequently, the landslide hazard would appear to depend solely on these landslides. In reality, this isn't the case. What actually happened in this time frame is that the small, shallow landslides reported in the multi-temporal inventory occurred paroxysmal and therefore left clearly visible traces on the landscape, easily mappable by aerial photographs. On the other hand, the evolution of medium-large, deep-*

*seated landslides, which is typically very slow, continuous, and/or seasonal, leaves no visible changing on the landscape and is therefore not mappable by aerial photographs. However, we know that medium- and large-scale landslides create the greatest hazard conditions in our study area. In particular, what we observe is that the LHZs (Landslide Hazard Zones) for medium-sized landslides with the highest spatial frequencies (although not referred to a defined time frame) are those with the most evident damages to structures and infrastructures and most dynamic PSs.*

*Basing on these evidences, we maintein it is more effective and realistic to determine the hazard based on the spatial frequency of all the landslides - even if measured over an indefinite time window - rather than limiting the hazard analysis to landslides from the multitemporal inventory, which ultimately are not representative of the landslide evolution of the area (except to a minimal extent).*

6. The manuscript introduces PS within LHZs "to identify active settlements," but the role of PS in the hazard inference remains vague (screening? validation? conditioning?). Spell out: (i) PS processing settings; (ii) deformation thresholds that define "active"; (iii) how PS density/velocity modifies frequency classes; (iv) how you treat layover/vegetation decorrelation; and (v) how PS evidence is integrated with inventory (e.g., Bayesian updating of activity state vs. a qualitative overlay). Without this, PS risks being read as a visual add-on rather than a quantitative component.

*6. Actually, we consciously decided not to include PS data as a quantitative component of our procedure. This is because it is known that PS data coverage across a territory is partial because it depends on land cover. This intrinsic characteristic of PS data is incompatible with our procedure, which, by definition, must necessarily produce a landslide hazard zoning using a homogeneous method across the entire territory, avoiding bias. However, where available, PS data represent an additional, updated, and independent source of information on the state of landslide activity that deserves to be used. We therefore chose to use PS data to complement the procedure's output. In other words, PS data cannot change the specific hazard (indicated in our procedure by a positional index) of a LHZ, but they add information on the state of landslide activity within an already classified LHZ. This additional information can be used by decision makers to determine the absolute hazard level of that territory, basing not only on the specific hazard provided by our procedure, but also on the PS data. In the new version of the manuscript, we will define the method more explicitly, including the role of PS in the hazard inference.*

*Regarding the points that the reviewer asks for clarification, it is specified that:*

*i) We used PS data already processed in the framework of the Doris EU project. The specifics of the processing of these data are available in the project reporting (https://cordis.europa.eu/project/id/242212/reporting) and in the published papers focusing on interferometry analysis related to the project, which are already cited in the manuscript (Bianchini et al., 2012 and 2015; Cigna et al., 2013; Raspini et al., 2016; Ciampalini et al., 2015);*

*ii) We consider as inactive ground motion values below the threshold of |2| mm/year. The value coincides with the minimum stability threshold used by Ciampalini et al., 2015 for the PSs of the entire Messina Province, which includes our study area. The value also is comparable to the lowest noise value that can be evaluated by observing different time series of ERS, ENVISAT and COSMO data randomly sampled in the study area;*

*iii) As previously mentioned, the PS data do not modify the frequency count resulting from our procedure;*

*iv) As mentioned before, we do not process PS data, but we use PS data already processed in the framework of the European Doris project, which are widely used and published in several scientific paper we have specifically cited (see above). However, we maintein that layover/vegetation decorrelation problems are negligible since: 1) as shown in Figure 13a, the examined PSs cover predominantly non-vegetated or subordinately poorly/sparsely vegetated areas; 2) the use of different PS data sources (ERS, ENVISAT, and COSMO) minimized decorrelation impact; 3) the study area is particularly suitable for relief visibility because it is characterized by a roughly N–S oriented valleys with E facing slopes, allowing for the detection of a great amount of radar benchmarks; 4) We complement hazard analysis with the countering of tens of PS data within individual LHZs and not basing on individual PS information. This type of slope scale analysis, allows us to overcome any localized decorrelation problems, which are therefore significantly mitigated. Overall, although we cannot exclude localized decorrelation problems, the above circumstances minimize their effects, which can thus be neglected. Finally, more in general, it is important to point out that PS data do not quantitatively impact the output of our procedure, thus the layover/vegetation decorrelation can be considered a negligible issue for the scope of the present work.*

*v) PSs are not quantitatively integrated into the inventory. In other words, landslides are not associated with velocity attributes derived from the PS analysis. In Figure 13b, we show a qualitative overlay between the PS data and our inventory map to show the good matching between the coherently moving PS clusters and the polygons - or parts of polygons - of many past landslides, while there is no evident association between the same clusters and the landslides of the multitemporal inventory. The evidence supports our thesis suggesting that it is sufficient to use past landslides as input data for the landslide hazard estimation procedure without necessarily having to carry out a multitemporal landslide inventory.*

7. The paper discusses reasoning behind frequency mapping but does not present a formal validation (e.g., chrono-validation, forward validation against a held-out recent period, or cross-regional transfer). I would consider some quantitative validation by some sort of chrono validation.

*As also suggested by the reviewer (in brackets) at the previous point, the PS data could be used as a semi-quantitative validation of the frequency associated with each LHZ. Indeed, according to our method, for the same level of intensity, higher frequency values determine conditions of greater hazard. Conditions of greater hazard are also determined by the landslides activity within each LHZ, which can be determined for the period 1992-2012 from the PS data. Therefore, only for the areas covered by PS data, the output of our procedure can be partially validated by verifying that: i) LHZs with a low frequency (equal to 1) are quiscient zones (i.e., maximum movements do not exceed 2 mm/year); ii) LHZs with a frequency greater than 1 are active zones (i.e., maximum movements exceed 2 mm/year). Visual inspection of Figure 12, expres this association between frequency values and landslide activity within LHZs. However, It is important to keep in mind that the absence of ongoing ground motion indicators within landslide polygons simply means that the landslide is quiescent, not that the hazard is low or negligible. Furthermore, more in generale, it is important to point out that PS data analysis can only be applied to slow-moving landslide phenomena. In the new version of the manuscript we will better describe the role and use of PS also as a validation tool.*

*A further validation tool can be based on the multi-temporal inventory. Indeed, the output of our procedure could be partially validated by verifying the presence (or the absence) of multi-*

*temporal landslides (occurred in the period 1955-2013) within the LHZs designed basing only on the occurrence of pre-1955 landslides. The information content of the multitemporal inventory can be particularly effective for validating the hazard of fast landslide, since even small fast landslide can have high magnitude (depending on the volume of the maximum expected boulder involved). This allows to complement the validation done with PS, which by definition cannot detect fast landslides. In the new version of the manuscript we will dedicate specific new text and a new figure to describe the two validation tools (multitemporal landslides and PS) of the output of our procedure.*

8. How would you assess the uncertainties in the hazard zones, and how better is it compared to geomorphic zonation by expert based analysis where expert looks at the geomorphology and defined the landslide zones. I think this needs to be discussed extensively.

*8. In our procedure, the Landslide Hazard Zones (LHZs) are defined as areas of possible (or probable) short-term evolution of existing landslides with similar characteristics (i.e. of type, volume, depth, and velocity). A LHZ is therefore a "landslide scenario" designed around a landslide or group of landslides and delimited using geomorphological criteria. Therefore, the uncertainty associated to each LHZ can be considered as coinciding with the difference between the area of the landslide polygon(s) within a given LHZ and the area of the LHZ itself. Basing on our dataset, this difference (the uncertainty) depends on the size of the landslides, and is greater the smaller the landslides. However, this excess area is not distributed equidistantly in all directions around the landslide(s) because it depends on the local morphology and the evolutionary trend of slope considering the (i) partial reactivation of portion of existing landslides through new landslides of younger generation, (ii) the lateral, head (retrogressive) or toe (progressive) expansion of the existing landslides, and (iii) the possible occurrence of new landslides of similar type and intensity. This is the main difference with the traditional geomorphological approach. In fact our procedure does not establish areas susceptible to landslides based on the geomorphologist's experience. Instead, our procedure defines the limits of the potential evolution of landslides or groups of landslides of an inventory map, primarily basing on the proximity and generational relationships identified between these landslides. We believe our approach limits operator (geomorphologist) subjectivity by constraint the choices underlying the hazard analysis to the data and metadata associated with a landslide inventory. This is a central point of our work. As a matter of fact, as indicated in the title and as shown throughout the manuscript, we apply the same procedure to inventory maps of varying detail, demonstrating the impact of the underlying data on the procedure's outcome, regardless of the operators (geomorphologists) and their experience. In the new version of the manuscript, we will reiterate this point and discuss it more extensively.*

---

## Author Comment (AC2)

***Responses to Reviewer #2***

Donnini et al. present a framework for estimating landslide hazard. As I understand, the framework differs from an earlier one in so far that it considers higher-resolution landslide inventories. In the way the study is presented, I struggle seeing the novelty and it seems quite technical, mostly comparing results to this earlier method, while the broader relevance and importance is unclear. Please see my main comments below.

*We thank Reviewer #2 for reviewing our article and for his helpful comments. As Reviewer #2 correctly writes, the framework for estimating landslide hazard presented in our MS builds on the method proposed by Cardinali et al. (2002), which we have acknowledged several times throughout the text, citing it as the "original method." However, differently from what the reviewer summarized, there are many differences that distinguish the framework proposed in our MS from the original method. The main differences are visually presented in Figure 3 and are: i) the analysis of three different inventories; ii) the comparison of the information content, materials, and methods used for the examined inventories with reference inventories; iii) a new method for estimating the frequency of all landslides (slow, fast, and rapid); iv) the use of the information derived from the ground motion time series as obtained through the persistent scatterers (PS) technique. These innovations certainly impact the final hazard assessment. Therefore, to visualize this impact, it is necessary to compare the results of the original and new methods. We acknowledge that this is a rather technical aspect of the MS, which we should nevertheless maintain with the aim of highlighting the improvement brought by our original results to the landslide hazard estimation. However, in the new version of the MS, we will strive to highlight the broader relevance and importance of our study, which currently seem partially hidden by the technical aspects.*

What exactly is the novelty of the study? It seems like it's an optimized version of earlier work. Many of the findings are compared to this earlier version of the method, but the broader relevance and implications are unclear. Findings like generational inventories (ie higher resolution) inventories provide more information is not surprising.

*The response to the previous point also largely addresses this comment. However, differently to what the reviewer claims, our findings do not stress the unsurprising circumstance that higher resolution inventories provide more information for landslide hazard assessment. Actually, in a certain sense, it's quite the opposite. In fact, at line 767-772, the text read: "The comparison of different inventories indicates that the contribution of multi-temporal landslides (i.e. the landslides recognized after 1955) only enriches the information relating to the smaller and more scattered LHZs. Instead, multi-temporal landslides do not contribute to characterizing the landslide hazard of the study area, which instead is mostly controlled by previous landslides, already present in the generational-historical inventory, and only partially, in the basic-historical inventory." In other words: the multi-temporal inventory - which is the most detailed inventory we compiled - it is not decisive for the definition of the landslide hazard. Our findings rather substain that is important to base landslide hazard analysis on generational-historical inventories that adequately characterize the complexity of landslide clusters compared to the basic-historical inventories. This conclusion - independently confirmed by ground motion data - is not trivial and raises questions and research needs on the reliability and availability of landslide inventory maps used for landslide hazard studies and modelling.*

As I understand, landslide inventories were generated, but the underlying mechanisms driving the hazard and leading to changes through time is hardly discussed. Understanding this would greatly help in assessing future hazards.

*Our method base on the distribution and pattern of landslides contained in the available inventories, which allow one to infer the possible evolution of slopes, in terms of the most probable type of failures, and their expected frequency of occurrence and intensity. These are the conditions driving the landslide hazard of an area. The areas of evolution of these landslides are here named Landslide Hazard Zones (LHZs), and are defined as areas of possible (or probable) short-term evolution of existing landslides with similar characteristics (i.e. of type, volume, depth, and velocity). As specified in the ancillary materials, a LHZ is a "landslide scenario" delimited using geomorphological criteria considering (i) the partial or total reactivation of existing landslides, (ii) the lateral, head (retrogressive) or toe (progressive) expansion of the existing landslides, and (iii) the possible occurrence of new landslides of similar type and intensity. Within each LHZ, the levels of landslide hazard are expressed using an index that conveys, in a simple and compact format, information on the landslide frequency and the landslide intensity. Importantly, different LHZs with different levels of hazard can be determined for each type of failure observed on an elementary slope (e.g. fast-moving rock falls, rapid-moving debris flows, slow-moving earth-flow slumps or compound failures). This is very important because - according to our procedure - the overall hazard of an elementary slope is given by the combination of as many specific hazards as there are classes of typology and intensity of landslides recognized on that slope. This condition allowing us to keep track of the hazard posed also by pre-existing landslides which - although often partially hidden or remodeled - can represent the underlying mechanisms driving the hazard of most recent, and often active landslides. For instance, slow and pre-existing landslides partially or entirely overlain by fast and recent ones is a common geomorphological pattern in the study area, well captured by our generational-hystorical inventory. In such contexts, the continuous - albeit slow - movement of the underlying landslides generates and maintains the conditions of instability that promote the frequent activation of rock-fall within extensive rock fall areas. While we maintain that our results illustrate this complex chain of risks, we agree with the reviewer that this evidence is poorly discussed, remaining obscured by the technical aspects of the MS. In the new version of the manuscript, we will ensure a better balance between the discussion of the technical aspects and the scientific implications.*

The Intorduction remains at a superficial level. Pros and cons of broad classes of methods (geomorphological, statistical, …) are listed, but with little detail and without identifiying the research gaps that are connected to this work. What is the motivation for the presented work?

*The main scientific advances presented in the recent landslide hazard literature concerns physically-based analysis and statistical/ML approaches. Our study, however, presents a semi-quantitative heuristic approach, which, unlike modeling approaches, is little explored in the scientific literature, and this is a good motivation for the present work. The method we propose does not express landslide probability in modeling terms (probabilistic or deterministic), but defines landslide hazard areas based on landslides that have already occurred and the potential evolution zones for individual landslides or groups of landslides. Within these areas, it defines the hazard level based on the magnitude, velocity and frequency of landslides that have already occurred and are expected. While highlighting certain limitations, we maintain that our method represents an important advance in the field of semi-quantitative landslide hazard estimation because: i) it utilizes and optimizes information on landslide intensity and recurrence contained in inventory maps; ii) it takes into account the spatial and temporal dimensions of hazard; iii) it establishes a logical and reproducible procedure; iv) it overcomes the subjectivity of traditional geomorphological approaches; v) it provides maps directly applicable to land-use planning. In the new version of the MS, we will appropriately edit and strengthen the introduction by presenting and arguing for these improvements brought by our study, with respect to the research gaps and critical issues implicit in other heuristic approaches. In addition, as also stated to the other reviewer, the presentation of a new method for landslide hazard assessments is not the only result of this*

*work. Indeed, as the title suggests, the main objective of this paper is to demonstrate and measure the effect of different mapping approaches on landslide hazard estimation. In the new version of the manuscript, we will better highlight this additional result and its scientific relevance.*